# An Animal Model for Chronic Meningeal Inflammation and Inflammatory Demyelination of the Cerebral Cortex

**DOI:** 10.3390/ijms241813893

**Published:** 2023-09-09

**Authors:** Lukas Simon Enz, Anne Winkler, Claudia Wrzos, Boris Dasen, Stefan Nessler, Christine Stadelmann, Nicole Schaeren-Wiemers

**Affiliations:** 1Neurobiology, Department of Biomedicine, University Hospital Basel, University Basel, Hebelstrasse 20, CH-4031 Basel, Switzerland; lukas.enz@unibas.ch; 2Institute of Neuropathology, University Medical Center Göttingen, 37075 Göttingen, Germany; anne.winkler2@fu-berlin.de (A.W.); stefan.nessler@googlemail.com (S.N.); cstadelmann@med.uni-goettingen.de (C.S.); 3Tissue Engineering, Department of Biomedicine, University Hospital Basel, University Basel, CH-4031 Basel, Switzerland; boris.dasen@unibas.ch

**Keywords:** demyelination, cortical inflammation, multiple sclerosis, grey matter lesion, MS cortex pathology

## Abstract

Modeling chronic cortical demyelination allows the study of long-lasting pathological changes observed in multiple sclerosis such as failure of remyelination, chronically disturbed functions of oligodendrocytes, neurons and astrocytes, brain atrophy and cognitive impairments. We aimed at generating an animal model for studying the consequences of chronic cortical demyelination and meningeal inflammation. To induce long-lasting cortical demyelination and chronic meningeal inflammation, we immunized female Lewis rats against myelin oligodendrocyte glycoprotein (MOG) and injected lentiviruses for continuing overexpression of the cytokines TNFα and IFNγ in the cortical brain parenchyma. Immunization with MOG and overexpression of TNFα and IFNγ led to widespread subpial demyelination and meningeal inflammation that were stable for at least 10 weeks. We demonstrate here that immunization with MOG is necessary for acute as well as chronic cortical demyelination. In addition, long-lasting overexpression of TNFα and IFNγ in the brain parenchyma is sufficient to induce chronic meningeal inflammation. Our model simulates key features of chronic cortical demyelination and inflammation, reminiscent of human multiple sclerosis pathology. This will allow molecular, cellular and functional investigations for a better understanding of the adaptation mechanisms of the cerebral cortex in multiple sclerosis.

## 1. Introduction

Multiple sclerosis (MS) is the most common chronic inflammatory and neurodegenerative disease of the central nervous system. The pathologic hallmark of MS is the formation of focal areas of myelin loss, termed lesions. Besides the commonly described white matter lesions, extensive grey matter lesions are found in MS cerebral cortex and deep grey matter [1,2,3]. Grey matter lesions are reported to occur in up to 90% of patients with chronic MS [4]. Early-stage MS grey matter lesions were shown to contain myelin-laden macrophages as well as perivascular CD3+ and CD8+ T-cell infiltration [5,6]. In contrast, chronic grey matter lesions characteristically lack significant infiltration of immune cells but show diffuse microglial activation [7,8]. In both the early and in the chronic stage, grey matter lesions were shown to be associated with meningeal inflammation [5,9]. Cortical demyelination also correlates with loss of neurons, axons, synapses, and glial cells [8,10,11,12,13]. Grey matter lesions further seemed to have a more efficient myelin repair and less gliosis compared to white matter lesions.

Cortical atrophy and demyelination in MS are associated with lymphoid aggregates in the meninges [8]. Further, higher levels of interferon gamma (IFNγ) and tumor necrosis factor alpha (TNFα) in the meninges and in the cerebrospinal fluid were identified [7,14]. Although grey matter pathology in MS has received rapidly growing interest over the past two decades, many aspects of the chronic pathological processes and the adaptations to a chronically demyelinated environment remain elusive.

Long-standing cortical demyelination may contribute to neuronal damage and degeneration. It is estimated that 40–70% of MS patients suffer from deficits of memory and attention, reduction of cognitive flexibility, and impairment of executive functions [15,16]. These symptoms are therapeutically very difficult to address and are mechanistically poorly understood, but it is strongly suspected that these cognitive symptoms are associated with changes in cortical grey matter such as demyelinated lesions and atrophy [14,17]. Besides the well-described demyelinated grey matter lesions, diffuse grey matter abnormalities in non-lesional normally myelinated areas, such as diffuse microglial activation, diffuse axonal injury, and astrogliosis, have also been observed [18,19,20,21].

To study the pathomechanisms and functional consequences of chronic grey matter demyelination and meningeal inflammation in detail, an experimental model replicating the key phenotypes is required. Such a model was established by Merkler et al. in 2006 [22] by immunizing Lewis rats with recombinant myelin oligodendrocyte glycoprotein (MOG) and incomplete Freund’s adjuvant (IFA) [6,22]. This animal model was characterized by highly reproducible focal cortical demyelination and inflammation, followed by rapid resolution and remyelination. This model, however, did not give rise to a chronic cortical pathology.

Based on our results, we report the establishment of an animal model for studying the consequence of chronic cortical grey matter demyelination and meningeal inflammation.

## 2. Results

### 2.1. Long-Lasting Cerebral Overexpression of TNFα and IFNγ

The DNA sequences encoding for *tnfa*, *ifng*, and *lacZ* were cloned into the pULTRA plasmid under the human ubiquitin C promoter to generate the lentiviral constructs LV-*tnfa*, LV-*ifng*, and, as a control, LV-*lacZ* (Figure 1A). ELISA quantification of lentiviral-associated P24 protein revealed an average of 8.2 × 10^6^ transfected units per ml over all batches and viruses. HEK293T cells were transduced, leading to efficient long-lasting overexpression of TNFα, IFNγ, and β-galactosidase. Biological activity of each batch of lentiviruses was confirmed by transducing HEK293T cells and subsequent qualitative ELISA measurements of TNFα and IFNγ protein expression from the supernatant. Controls were animals that were MOG-immunized and transduced with LV-*lacZ* to overexpress β-galactosidase, MOG-immunized animals with a single injection of recombinant TNFα and IFNγ proteins, and IFA-immunized animals transduced with LV-*tnfa* and LV-*ifng*. Animals immunized with MOG and transduced with LV-*tnfa* and LV-*ifng* represent the experimental model (Figure 1B). The rats were perfused 5 days, 4 weeks (28 days), 7 weeks (49 days) and 10 weeks (70 days) after injection. Immunohistochemical staining for GFP in animals chronically overexpressing TNFα and IFNγ revealed eGFP expression in cells suggestive of neuronal and glial cells at each time point (Figure 1D,E). The GFP-positive cells and the marker dye monastral blue were detected in a trajectory along the injection route from neuronal layers I to VI, sometimes reaching into the cingulate fiber bundle in the cortex (Figure 1D).

### 2.2. Chronic Overexpression of TNFα and IFNγ Is Required for Meningeal Inflammation

Five days after stereotactic injection, a widespread meningeal infiltrate was identified in all animals chronically overexpressing TNFα and IFNγ, irrespective of MOG immunization (Figure A1A–D, arrows). The mean maximum thickness of meninges was 99.1 μm in animals chronically overexpressing TNFα and IFNγ compared to 16.8 μm in animals injected with recombinant TNFα and IFNγ (cytokines) or with LV-*lacZ* lentivirus. The meningeal infiltrate always involved the longitudinal fissure and spread laterally 5.1 mm on average in MOG-immunized and chronically overexpressing TNFα and IFNγ animals and 4.2 mm in non-immunized, chronically overexpressing TNFα and IFNγ animals (Figure A1G). In contrast, there was no infiltration of the meninges with ED1+ cells observed in animals injected with cytokines or LV-*lacZ* lentivirus (Figure A1A,B,G).

Four weeks after LV-*tnfa* and LV-*ifng* injection, there was still no significant difference between MOG-immunized (Figure A1H) and non-immunized animals and animals injected with recombinant cytokines still did not show any meningeal inflammation. One of three animals injected with LV-*lacZ* showed a mild meningeal inflammation less than in MOG-immunized animals chronically overexpressing TNFα and IFNγ.

Ten weeks after LV-*tnfa* and LV-*ifng* injection, the meningeal infiltrate was significantly larger in MOG-immunized animals (Figure A1I) compared to non-immunized animals. In animals chronically overexpressing TNFα and IFNγ, the size of the meningeal infiltrate did not significantly change over time neither in MOG-immunized animals (F(3,18) = 1.357, *p*-value = 0.288, Figure A1J) nor in non-immunized animals (F(2,11) = 2.195, *p*-value = 0.158, Figure A1K).

### 2.3. Chronic Overexpression of TNFα and IFNγ Leads to a Stable T- and B-Cell Meningeal Infiltrate for at Least 10 Weeks in MOG-Immunized Animals

The meningeal-infiltrating cells comprised macrophages, T-cells, and B-cells (Figure 2 and Figure 3) with a ratio of T- to B-cells of approximately 2:1. Five days after injection, the number of infiltrating cells in meninges of MOG-immunized animals chronically overexpressing TNFα and IFNγ was on average 1826 CD3+ cells (Figure 2D) and 1000 CD45R+ cells per coronary histological section (Figure 3D). This was comparable to 1865 CD3+ cells and 690 CD45+ cells detected in non-immunized animals chronically overexpressing TNFα and IFNγ. In animals injected with recombinant cytokines or with LV-*lacZ*, the number of infiltrating CD3+ cells was on average 137 (cytokines) or 82 (LV-*lacZ*), which was significant. The numbers of CD45R+ cells were on average 138 (cytokines) or 155 (LV-*lacZ*).

Four weeks after the injection, the number of CD3+ cells was on average 817 (Figure 2E) and for CD45R+ cells 705 (Figure 3E) in MOG-immunized, chronically overexpressing TNFα and IFNγ animals but no significant difference was observed compared to non-immunized animals. In cytokine-injected and in LV-*lacZ* animals, the average numbers of CD3+ and CD45R+ cells were at very low levels. After 10 weeks, there was no significant difference between MOG-immunized animals chronically overexpressing TNFα and IFNγ, showing numbers of 1056 CD3+ (Figure 2F) and 77 CD45R+ (Figure 3F), compared to non-immunized animals (554 vs. 22).

In MOG-immunized animals chronically overexpressing TNFα and IFNγ, analysis of variance showed no significant decline of CD3+ cell infiltrate (F(3,18) = 1.977, *p*-value = 0.154, Figure 2G) or of CD45R+ cell infiltrate (F(3,18) = 2.233, *p*-value = 0.119, Figure 3G), indicating a meningeal inflammation over at least 10 weeks. In non-immunized animals, analysis of variance did not show a significant decline of CD3+ cell infiltrate (F(2,11) = 2.326, *p*-value = 0.144, Figure 2H) but it showed a significant decrease in the CD45R+ meningeal infiltrate over time (F(2,11) = 6.352, *p*-value = 0.015, Figure 3H). Tukey post hoc analysis revealed a significant decrease in the number of CD45+ cells from five days to ten weeks (adj. *p*-value = 0.011).

### 2.4. CD3+ and CD45R+ Parenchyma Infiltration Is Present in All Study Groups at the Site of Injection but Decreases over Time

Five days post injection, we detected infiltrating CD3+ and CD45R+ cells in proximity to the injection site (Figure 2B for CD3+ cells, Figure 3B for CD45R+ cells). This infiltrate was largest in MOG-immunized animals chronically overexpressing TNFα and IFNγ with a mean of 1118 CD3+ cells (Figure 2I) and a mean of 267 CD45R+ cells (Figure 2I) per coronary histological section. There was no significant difference compared to any other group (Figure 2).

Four weeks post injection, the infiltrated cells were still present in animals that were MOG-immunized and chronically overexpressing TNFα and IFNγ with a mean of 475 CD3+ cells per coronary histological section (Figure 2J) and of 243 CD45R+ cells (Figure 3J). Non-immunized animals chronically overexpressing TNFα and IFNγ showed on average 269 CD3+ cells and 63 CD45R+. In cytokine-injected animals, the infiltrate had already decreased more rapidly with an average of 106 CD3+ and 22 CD45R+ cells, that was comparable to LV-*lacZ*-injected animals that had on average 113 CD3+ and 47 CD45R+ cells per coronary histological section (Figure 2J and Figure 3J).

After 10 weeks, the numbers of infiltrating cells were still comparable in MOG-immunized and TNFα and IFNγ overexpressing animals, that had a mean of 103 CD3+ cells (Figure 2K) and only 4 CD45R+ cells (Figure 3K), and non-immunized animals with 199 CD3+ cells and 36 CD45R+.

Analysis of variance of MOG-immunized animals overexpressing TNFα and IFNγ revealed a significant decrease over time in the CD3+ and CD45R+ parenchymal infiltration (CD3+: F(3,18) = 10.6, p < 0.001, Figure 2L, CD45R+: F(3,18) = 3.029, *p*-value = 0.056, Figure 3L). Tukey post hoc analysis revealed a decrease in CD3+ cell numbers between five days and all later time points. Analysis of non-immunized animals also revealed a significant decrease over time (CD3+: F(2,11) = 7.775, *p*-value = 0.008, Figure 2M; CD45R+: F(2,11) = 14.39, *p*-value < 0.001, Figure 3M). Tukey post hoc analysis revealed a decrease between five days and all later time points.

### 2.5. Blood–Brain Barrier Leakage Is Present for at Least 10 Weeks Post Injection

To analyze blood–brain barrier breakdown, we injected the second cohort of rats with FITC-labeled albumin prior to perfusion and performed an anti-FITC staining. This revealed an average of 0.01 mm^2^ of stained area over all time points for MOG-immunized animals chronically overexpressing TNFα and IFNγ (Figure A2A). The leakage was present until 10 weeks post injection and there was no significant difference between the time points (Figure A2B). The non-immunized animals chronically overexpressing TNFα and IFNγ showed an average FITC-albumin-stained area of 0.02 mm^2^. The leakage was present until 10 weeks post injection and we did not detect a significant difference between the time points (Figure A2C).

### 2.6. MOG Immunization Is Necessary for Cortical Demyelination

Five days after the stereotactical injection, staining for MBP expression revealed cortical demyelination in all study groups previously immunized against MOG (Figure 4A–D,G). All cortical demyelination detected was subpial, stretching from the midline laterally, predominantly on the injected hemisphere but also including the contralateral hemisphere (Figure 4F). The demyelinated area ranged from 0–19.5% of the total cortical area (Figure 4G). The percentage of demyelinated cortex was largest in animals chronically overexpressing TNFα and IFNγ with a mean of 13.3%, followed by animals injected once with the cytokines with a mean of 5.6% and animals injected with LV-*lacZ* with a mean of 4.8%. The non-immunized animals with chronic overexpression of TNFα and IFNγ did not show any demyelination.

Four weeks after the stereotactical injection, the subpial demyelination was still largest in MOG-immunized animals with chronic overexpression of TNFα and IFNγ with a mean of 5.5% (Figure 4H). Residual subpial demyelination was detected in animals injected with the recombinant proteins after four weeks with a mean of 0.9% and in animals injected with LV-*LacZ* with a mean of 1.3%. The non-immunized animals with chronic expression of TNFα and IFNγ did not show any demyelination.

After 10 weeks, the area of demyelination was still 5.5% in MOG-immunized animals with chronic overexpression of TNFα and IFNγ (Figure 4I). The non-immunized animals with chronic TNFα and IFNγ overexpression did not show any demyelination, indicating that MOG immunization is required for demyelination and that long-term intracortical overexpression of TNFα and IFNγ alone is not sufficient to cause demyelination.

The cortical demyelination in MOG-immunized animals with chronic expression of TNFα and IFNγ peaked early at five days post injection (Figure 4J) and decreased thereafter until four weeks post injection, suggesting partial remyelination took place. The area of demyelination then remained stable until 7 and 10 weeks post injection.

Remarkably, at all time points at least one animal showed demyelination in both hemispheres including two out of five animals even after 10 weeks (Figure 4E). Therefore, we conclude that subpial demyelination peaks early and is long-lasting in MOG-immunized animals with chronic LV-mediated TNFα and IFNγ expression.

### 2.7. Lack of Oligodendrocyte Precursor Proliferation Coincides with Chronic Demyelination

Intrigued by the chronic failure of complete remyelination subsequent to chronic expression of TNFα and IFNγ, we investigated the oligodendrocyte population by staining all oligodendrocyte lineage cells with OLIG2, mature myelin-maintaining oligodendrocyte precursor cells with anti-P25, and pre-respectively actively myelinating oligodendrocytes with BCAS1 [23]. Visual investigation did not reveal any distinct pattern in the analyzed area; in particular, we did not detect a rim of proliferation surrounding the demyelinated lesion. We counted the number of positive cells in a 0.75 mm^2^ area, located in the subpial cortex above the injection site, where demyelination was most pronounced (Figure A3). To better compare between the animals, we normalized the ipsilateral to the contralateral side and took the logarithm to centralize the ratio around zero. The *p*-values were calculated with a paired Welch two-sided *t*-test of the log-transformed data.

After five days, in MOG-immunized animals with chronic expression of TNFα and IFNγ, we detected a significant increase in the ratio of ipsilateral compared to contralateral OLIG2-positive cells (df = 5, *p*-value = 0.001, Figure A3A,D), P25-positive cells (df = 5, *p*-value = 0.35, Figure A3B,E), and BCAS1-positive cells (df = 5, *p*-value = 0.035, Figure A3C,F). We did not detect a significant difference in any of the other studied groups.

After four weeks, we detected an increased number of OLIG2 cells ipsilaterally compared to contralaterally in the MOG-immunized rats chronically expressing TNFα and IFNγ (df = 6, *p*-value = 0.008) and in non-immunized animals with chronic expression of TNFα and IFNγ (df = 3, *p*-value = 0.029). We further detected a significant increase in the ratio of ipsilateral to contralateral BCAS1-positive cells in the cytokine-injected animals (df = 2, *p*-value = 0.025). After 10 weeks, there was no significant difference between the ipsi- and contralateral hemispheres.

### 2.8. Demyelination Correlates with the Meningeal CD3+ Inflammation but Not with the Parenchymal Inflammation in MOG-Immunized and TNFα- and IFNγ-Overexpressing Animals

The observation that subpial demyelination correlates with meningeal inflammation in MS has led to the speculation of a mechanistic link between the two phenomena. Investigation of our animal model showed a significant correlation between the area of demyelinated cortex and the number of meningeal CD3+ cells (Figure A4A) in the group immunized against MOG and chronically overexpressing TNFα and IFNγ. The correlation with meningeal CD45R+ cell count did not reach significance (Figure A4B). Similarly, we did not detect a significant correlation between the proportion of demyelinated cortex and parenchymal CD3+ cell count (Figure A4C) and parenchymal CD45R+ count (Figure A4D).

## 3. Discussion

Our results demonstrate that continuing overexpression of TNFα and IFNγ generated by lentiviral intracortical injection of MOG-immunized rats leads to chronic demyelination and meningeal inflammation.

In a previously described animal model, MOG-immunized rats were injected with recombinant proteins TNFα and IFNγ, giving rise to subpial and intracortical demyelinated lesions with infiltrating immune cells, complement deposition, acute axonal damage, and neuronal cell death, followed by a rapid resolution of the lesions [22]. A less traumatic version of this model was achieved by injecting the recombinant proteins TNFα and IFNγ into the subarachnoidal space. This approach gave rise to a similar pathology to the intracortical injections [7]. These models, using recombinant proteins, demonstrated that immunization against MOG and the injection of both TNFα and IFNγ are required for demyelination. In our model, we demonstrate that immunization against MOG is necessary for demyelination. Even a 10-week overexpression of TNFα and IFNγ alone did not lead to demyelination. Demyelination thus cannot be a direct effect of cytokine expression and meningeal inflammation but, in our model, also requires a priming of the immune system against a component of myelin.

The lesions were largest in animals, which were immunized with MOG and chronically overexpressing TNFα and IFNγ. However, the control virus leading to chronic β-galactosidase expression also gave rise to subpial demyelination comparable to the injection of the recombinant proteins TNFα and IFNγ. This could be a consequence of the direct traumatic tissue damage caused by the intraparenchymal injection, leading to an opening of the blood–brain barrier. Alternatively, it could be a consequence of lentiviral particles being injected into the parenchyma or β-galactosidase provoking an immune response. Merkler et al. found only limited demyelination when injecting phosphate-buffered saline into MOG-immunized animals instead of the recombinant proteins TNFα and IFNγ [22], suggesting that both proposed mechanisms might play a role in the animal model shown here.

It has further been demonstrated that one cytokine alone is not sufficient to generate widespread acute cortical demyelination [7,22]. We have demonstrated that the simultaneous injection of both LV-*tnfa* and LV-*infg* leads to demyelination in MOG-immunized animals, but we did not test the lentiviruses alone. It is not evident that both cytokines together are necessary and one alone may be sufficient to initiate and uphold the demyelination and inflammation.

The previous acute models recovered within two to four weeks. In comparison, our model causes stable subpial demyelination for at least 10 weeks and possibly much longer. In another approach by [24], a catheter was installed at the border of the white and cortical grey matter of MOG-immunized animals, allowing multiple injections of recombinant protein TNFα and IFNγ that generate demyelinating lesions for at least four weeks. This approach, however, causes direct trauma to the brain parenchyma, potentially greater than the injection of the lentivirus. It depends, in addition, on the permanent installation of foreign material, penetrating the skin, skull, and blood–brain barrier.

Further, we have shown that the meningeal inflammation depends only on the chronic overexpression of TNFα and IFNγ, as described before in the acute models [7]. The parenchymal infiltration detected was primarily limited to the injection site and occurred irrespective of the type of injection. It decreased in all groups over time.

After an initial phase with partial remyelination between five days and four weeks post injection, the extent of demyelinated lesions, the meningeal inflammation, and the number of oligodendrocyte lineage cells were stable until at least ten weeks post injection. The previous model with recombinant cytokine injection led to a rapid remyelination, even after repeated acute demyelination [25]. In our model, complete remyelination might fail directly due to the chronic release of TNFα and IFNγ or indirectly by effects inflicted by the meningeal inflammation. The density of oligodendrocyte lineage cells did not decrease in our model. We speculate that in the initial phase a certain loss of oligodendrocytes and a reactive proliferation of oligodendrocyte precursor cells occur. To support this, both TNFα and IFNγ have previously been shown to lead to oligodendrocyte death and demyelination [26,27]. However, we might have overlooked this due to the temporal resolution of our analysis. We could not demonstrate any significant changes in the overall number of oligodendrocytes or oligodendrocyte precursor cells.

In a parallel study, a very similar model has been described [28]. The authors immunized Dark Agouti rats against MOG and injected lentiviruses to overexpress TNFα and IFNγ in the subarachnoidal space, efficiently transducing meningocytes, in contrast to the intracerebral injections in the animal model described here. Their approach gave rise to bihemispheric, symmetric meningeal inflammation and subpial demyelination, closely resembling aspects of human cortical MS pathology. While the changes described here are very similar, we thus see predominantly unilateral pathology. This is an advantage, enabling us to use the weakly affected contralateral hemisphere as an internal control. We see our approach as an alternate version of the model suggested by James et al. [28].

In MS research, lesions spanning the whole cortical thickness from the subpial area to the white matter, termed type III cortical lesions, have been shown to be the most common type of cortical MS lesions and have been demonstrated to be unique to MS pathology [29,30]. Interestingly, the acute models described previously and the model established here lead to a subpial demyelination pattern as seen in MS. While Merkler et al. discussed the drainage of the recombinant proteins along predetermined anatomical routes from the white matter to the meninges as a possible cause of this pattern [22], Gardner et al. suggested that cytokines produced from the meningeal infiltrates may diffuse into the brain parenchyma and cause a subpial pattern of demyelination [7]. In our model, both mechanisms are possible: a direct effect by the lentivirus, leading to chronic overexpression of TNFα and INFγ, or an indirect effect caused by cytokine production from the meningeal infiltrates. In MS, an association between subpial demyelination and meningeal inflammation has been discussed [8,9,18], but the nature of any causal connection between the two phenomena remains elusive. Analysis of our data revealed a significant correlation between the demyelinated fraction of the cortex and the meningeal inflammation, whereas the parenchymal infiltration did not show such a correlation. However, we cannot conclude that meningeal inflammation is necessary for the development of subpial demyelination from our model.

## 4. Materials and Methods

### 4.1. Subcloning of Lentiviral Plasmids

The lentiviral plasmid pULTRA was used as s backbone and was a gift from Malcolm Moore (Addgene plasmid #24129, Addgene, Watertown, MA, USA) [31]. The sequence for *tnfa* (reference sequence: NM_012675.3, nucleotides 5′-145-852-3′) was acquired from rat cDNA. The sequence for *ifng* (reference sequence: NM_138880.2, nucleotides 5′-10-480-3′) was amplified by polymerase chain reaction (PCR) from a cDNA ORF clone plasmid (RG80234-UT, Sino Biological, Wayne, PA, USA). Primers for the PCR of *tnfa* and *ifng* were designed with specific restriction enzyme sequences at the 5′ and 3′ sites (Figure 1A). The PCR products were separated by gel electrophoresis and the specific band purified by a gel DNA recovery kit according to the manufacturer’s protocol (D4007, Zymo Irvine, CA, USA). The sequence for *lacZ* was obtained by restriction digest with BamHI (R3136S, New England Biolabs, Ipswich, MA, USA) of the plasmid LV-*lacZ* received as a gift from Inder Verma (Addgene plasmid #12108) [32]. Restriction digests were performed in CutSmart buffer (B7204S, New England Biolabs, Ipswich, MA, USA), using 6 units of restriction enzyme per microgram of cDNA at 37 °C for 3 h. cDNAs were subcloned into the pULTRA backbone with the Rapid DNA Dephos & Ligation Kit (4898117001, Hoffmann-La Roche, Basel, CH, Switzerland) according to the manufacturer’s protocol. Amplification of the plasmids was carried out in Stbl3 bacteria (C737303, Thermo Fisher Scientific, Waltham, MA, USA). The plasmids were thereafter purified with the miniprep classic kit (D4015, Zymo Irvine, CA, USA) and sent to Microsynth (Balgach, Switzerland) for sequencing.

### 4.2. HEK293T Cells

HEK293T cells were used for lentivirus production and lentiviral quality control.

### 4.3. Lentivirus Production

Lentiviruses containing the sequence for enhanced green fluorescent protein (eGFP) with either *lacZ* (LV-*lacz*), *tnfa* (LV-*tnfa*), or *ifng* (LV-*ifng*) were generated in HEK293T cells. The lentiviral helper plasmids pMD2.VSV-G, (Addgene plasmid #12259), pMDLg/pRRE (Addgene plasmid #12251), and pRSV-Rev (Addgene plasmid #12253) were a gift from Didier Trono (Addgene, Watertown, MA, USA) [33]. For a T75 flask with 70% confluent HEK293T cells, a 1.5 mL tube of Opti-MEM medium (31985062, Thermo Fisher Scientific) was mixed with 5 μg of each helper plasmid and 6.5 μg of the lentiviral plasmid. Further, one 15 mL tube with 0.75 mL Opti-MEM medium was mixed with 60 μL Lipofectamine 2000 Transfection Reagent (No. 11668027, Thermo Fisher Scientific). The diluted plasmids were mixed with the Lipofectamine solution, vortexed, and incubated for 5 min at room temperature. The DNA–lipid complex was then added to the cells and incubated for 18 h, removed, and replaced with 7 mL fresh HEK293T culture medium. After 48 h, the cell culture medium was harvested, transferred to a 15 mL tube, and centrifuged for 15 min at 3000× *g* to pellet cells and debris. The supernatant was then filtered through a 0.45 μm filter on a syringe. To concentrate the viral particles, 2.3 mL Lenti-X-concentrator was added before processing according to the manufacturer’s protocol (631232, Takara, Kyoto, Japan). The lentiviral particles were resuspended in PBS and aliquoted and stored at −80 °C.

### 4.4. Lentivirus Quantification

Concentration of lentiviral particles was determined by estimating the concentration of lentivirus-associated p24 protein with the QuickTiter Lentivirus Titer Kit according to the manufacturer’s protocol (VPK-107, Cell Biolabs, Inc., San Diego, CA, USA). For calculating the approximate number of transfected units, the conservative assumption of 1000 P24 proteins per lentiviral particle was applied (range suggested by the manufacturer: 100–1000).

### 4.5. Lentivirus Quality Control

To confirm the production of functional lentiviruses, production of TNFα and IFNγ was verified by enzyme-linked immunosorbent assay (ELISA). HEK293T cells were seeded into 6-well plates at a density of 200,000 cells per well and cultivated for 18 h before transduction with estimated viral particles per cell of either LV-*tnfa* or LV-*ifng*. The cells were transferred to fresh flasks and the medium was exchanged three times per week. After one week, the supernatant and the cells were assumed free of active lentiviral particles. Cell culture medium was then replaced with 1 mL fresh medium without serum and 24 h later ELISA was performed. For TNFα and IFNγ ELISA, an unlabeled TNFα capture antibody (dilution 1:167, 506102, BioLegend, San Diego, CA, USA) or an unlabeled IFNγ capture antibody (diluted 1:500, 507802, BioLegend), respectively, was diluted in coating buffer (421701, BioLegend) and 100 μL per well was distributed in a 96-well plate (423501, Biolegend). After 18 h of incubation at 4 °C, wells were washed with 0.05% Tween 20 in PBS three times and 200 μL blocking solution (421203, BioLegend) was added per well for one hour. Wells were then washed and incubated for three hours with 100 μL pure or 1:100 diluted supernatants from the HEK293 cell culture transfected with LV-*tnfa* or LV-*ifng*. Wells were washed and the biotin-labeled detection antibody against TNFα (diluted 1:1000, 516,003, BioLegend, San Diego, CA, USA) or IFNγ (diluted 1:500, 518,803, BioLegend, San Diego, CA, USA), respectively, was added for one hour. Wells were washed and 100 μL per well of avidin-horseradish peroxidase conjugate was added (diluted 1:1000 in blocking buffer, 405103, BioLegend). Thereafter, wells were washed, and 200 μL substrate solution (421101, BioLegend) was added per well. Upon sufficient color development, 100 μL stop solution (423001, BioLegend) was added. Absorption was read at 450 nm.

### 4.6. Animals

Adult female inbred Lewis rats (185–250 g) were purchased from Janvier Labs (Saint-Berthevin, France) and used as described before [34]. Animal housing and all animal experiments were performed at the experimental animal facility of the University Medical Center Göttingen, Germany. The rats were housed in cages with up to five animals each under constant temperature and humidity, a 12/12 h light/dark cycle, and with free access to food and water. All animal experiments were performed in accordance with the European Communities Council Directive of 22 September 2010 (2010/63/EU) and were approved by the Government of Lower Saxony, Germany.

### 4.7. Immunization and Intracerebral Stereotactic Injection

The immunization and intracerebral stereotactic injection were performed as described before [22]. Eighteen days after immunization, intracerebral stereotactic injection was performed as described [22]. For this, rats were anesthetized by intraperitoneal injection of ketamine/xylazine and fixed in a stereotactic frame. A one-centimeter incision of the skin on the midline of the skull was made to expose the bregma and a hole was drilled 2 mm lateral and 1 mm caudal of the bregma. Using a finely calibrated glass capillary, 2.5 μL solution was carefully injected into the cortex, containing either LV-*tnfa* and LV-*ifng* mixed 1:1, LV-*lacZ*, or the recombinant cytokines TNFα (250 ng; P16599, R&D Systems, Abingdon, UK) and IFNγ (150 U; 400–20, PeproTech, London, UK). Monastral blue was added to mark the injection site. After the injection, the glass capillary was retracted and the skin was closed with a suture [34]. Rats were perfused after 5 days, 4 weeks, 7 weeks, or 10 weeks to investigate lesion evolution. For this, the animals were anesthetized with isoflurane, receiving a lethal dose of ketamine/xylazine and perfused transcardially with PBS followed by 4% paraformaldehyde. Subsequently, the heads were postfixed in 4% paraformaldehyde at 4 °C for two days before removing the brain and embedding it in paraffin.

### 4.8. Composition of Study Groups

In accordance with the 3R principle to reduce, refine, and replace animal experimentation, not all groups were analyzed at all time points, but the experiments were performed with four different models at four time points to address specific questions (Figure 1B). Animals immunized with MOG and transduced with LV-*tnfa* and LV-*ifng* represent the experimental model and were studied at 5, 28, 49, and 70 days to assess the development over time. This was compared with the previously published cytokine-induced experimental grey matter lesion model using MOG-immunized and recombinant TNFα- and IFNγ-injected animals at 5 days (peak of inflammation and demyelination in cytokine model) and 28 days (restitutio ad integrum in cytokine model) [21]. Also, we compared this experimental model to MOG-immunized animals transduced with LV-*lacZ* as a negative control at the time points 5 days and 28 days to control for early and late effects of the viral transduction. In addition, we compared the experimental group with non-immunized animals injected with LV-*tnfa* and LV-*ifng* to control for the MOG immunization at the time points 5, 28, and 70 days.

### 4.9. Enzyme-Linked Immunosorbent Assay for MOG Titers

For the measurement of the serum anti-MOG antibody titers, an ELISA was performed as described before [22]. Values two-fold above the background levels were considered positive.

### 4.10. Immunohistochemistry

Immunohistology was performed on 3 μm thin, formalin-fixed, paraffin-embedded (FFPE) coronary sections using antibodies for macrophages/activated microglia (CD68, clone ED1, Bio-Rad AbD Serotec, Oxford, UK), CD3+ T-cells (clone CD3-12Bio-Rad AbD Serotec, Oxford, UK), CD45R+ B-cells (clone HIS24, BD Biosciences, Franklin Lakes, NJ, USA), myelin basic protein (MBP, Abcam, Cambridge, UK), anti-neuronal nuclei (NeuN, Abcam, Cambridge, UK), oligodendrocyte lineage factor 2 (OLIG2, clone 211F1.1, Merck Millipore, Darmstadt, Germany), mature myelin-maintaining oligodendrocytes (P25/TPPP, clone EPR3316, Abcam, Cambridge, UK), pre-resp. actively myelinating oligodendrocytes (BCAS1, Santa Cruz, Heidelberg, Germany), FITC (HRP conjugated, Dako Deutschland GmbH, Hamburg, Germany), and green fluorescent protein (GFP, clone 6AT316, Abcam, Cambridge, UK).

### 4.11. Image Analysis

All stained tissue sections were scanned with an Olympus VS120-L100-W scanner fitted with a VC50 camera (Olympus, Shinjuku, Tokyo, Japan) with a 200× magnification at a resolution of 0.345 micrometers per pixel. For CD3 and CD45R staining, the cells were counted manually using NIS-Elements software (Version 5.11.00, Nikon, Tokyo, Japan). For demyelination, areas were delineated manually using CellSense software (Version 1.6, Olympus, Tokyo, Japan). For OLIG2, BCAS1, and P25 analysis, areas of 1 mm by 0.75 mm (0.75 mm^2^) were manually delineated in Adobe Photoshop (Version 22.1.0, Adobe Inc., San José, CA, USA) subpially above the injection site, where the pathology was most pronounced. OLIG2 and P25 were then automatically processed with Fiji (image processing package including ImageJ, Version 1.53a, Wayne Rasband, National Institutes of Health, Bethesda, MD, USA [35]), using the count objects function with the parameters: hue 106–189 (OLIG2) or 45–220 (P25), saturation 37–255 (OLIG2) or 80–255 (P25), brightness 0–255 (OLIG2) or 0–180 (P25), and size 11.9–238 µm^2^. Counted objects were marked and saved and all images were visually controlled for accurate counting. BCAS1-positive cells were counted manually using Adobe Photoshop (Version 24.7.0, Adobe, San José, CA, USA).

### 4.12. Statistical Analysis

Statistical analysis was performed with R. To compare means of two groups, two-sided Welch *t*-tests were performed and the degrees of freedom (df) and the *p*-values are reported. For comparisons of multiple time points, an analysis of variance (ANOVA) with subsequent Tukey post hoc correction was performed and adjusted *p*-values, the between-group and in-group degrees of freedom, and the F-value are reported as follows: F (between-group, in-group) = F-value. If the ANOVA test did not reach significance, the Tukey post hoc results are not reported. For correlations, Pearson correlations were calculated, and the correlation coefficient and the *p*-value are reported. For the analysis of the oligodendrocyte lineage cells (OLIG2, P25, BCAS1), the data were normalized by dividing by the number of cells in the same area in the contralateral hemisphere. Data were then log-transformed with base 10. Paired two-sided Welch *t*-tests were used to test for statistically significant changes between the ipsi- and contralateral cortex. For all tests, a *p*-value or adjusted *p*-value below 0.05 was considered significant.

## 5. Conclusions

We hope that in the future our model leads to new insights of the different contributors to MS cortical pathology. Chronic adaptations of the neuronal network and the supporting glial cells are some of the possibilities, as well as their influence on remyelination in the context of the adjacent inflamed meninges.

## Figures and Tables

**Figure 1 ijms-24-13893-f001:**
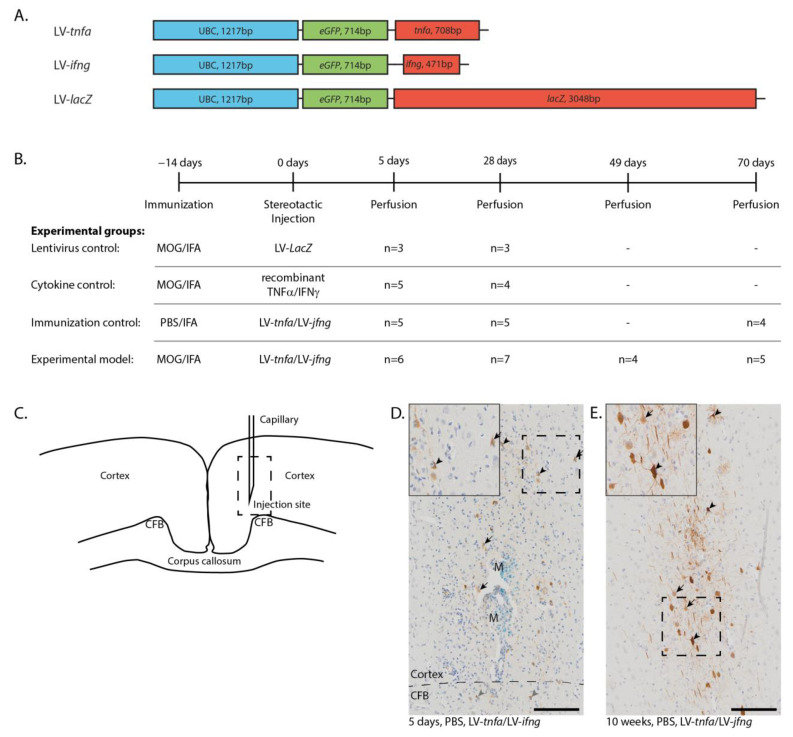
Lentiviral constructs and experimental design. (**A**) Schematic drawing of the lentiviral constructs depicting the human ubiquitin C promoter (UBC) and the sequences for enhanced green fluorescent protein (eGFP), tumor necrosis factor alpha (*tnfa*), interferon gamma (*ifng*), and beta galactosidase (*LacZ*). (**B**) Experimental timeline and table depicting the four experimental groups with the respective number of animals analyzed per study group and time point. Animals of the first two time points are pooled from two independent experiments. (**C**) Sketch of the injection site. Dashed box: approximate location of the images shown in (**D**,**E**). (**D**) Anti-GFP staining five days after the LV-*tnfa*/LV-*ifng* injection reveals cells of neuronal (arrows) and glial morphology in the cortex (black arrowheads) and in the cingulate fiber bundle (CFB, grey arrowheads). The tracer monastral blue (M) is detected along the injection trajectory. Dashed box: area of inset. Inset is at 160% magnification. (**E**) Anti-GFP staining 10 weeks after the LV-*tnfa*/LV-*ifng* injection shows cells of neuronal (arrows) and glial (arrowheads) morphology. Dashed box: area of inset. Inset is at 160% magnification. Scale bars: 100 μm. Abbreviations: MOG: recombinant myelin-oligodendrocyte glycoprotein, PBS: phosphate-buffered saline, IFA: incomplete Freund’s adjuvant, CFB: cingulate fiber bundle, M: monastral blue.

**Figure 2 ijms-24-13893-f002:**
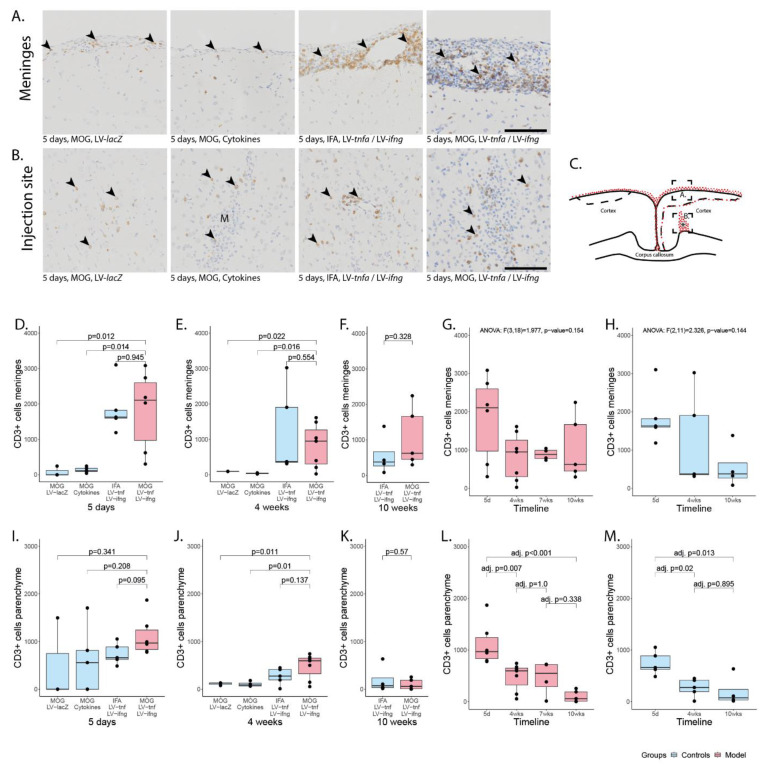
Meningeal and parenchymal infiltration with CD3-positive T-cells. (**A**) Representative images of CD3+ T-cells in meninges (arrowheads) and (**B**) in parenchyma (arrowheads). (**C**) Representative illustration of the inflammation pattern (red dots) five days after intracortical injection (asterisk), and delineation of the area of which images in A and B were taken (dotted squares). (**D**–**F**) Quantification of CD3+ T-cells detected in meninges for all study groups at five days, four weeks, and ten weeks after lentiviral injection. (**G**) Timeline of data presented in (**D**–**F**) of MOG-immunized, LV-*tnfa*/LV-*ifng*-injected animals and (**H**) of non-immunized, LV-*tnfa*/LV-*ifng*-injected animals. (**I**–**K**) Quantification of CD3+ T-cells detected in the parenchyma for all study groups five days, four weeks, or ten weeks after lentiviral injection. (**L**) Timeline of data presented in (**I**–**K**) of MOG-immunized, LV-*tnfa*/LV-*ifng*-injected animals and (**M**) of non-immunized, LV-*tnfa*- and LV-*ifng*-injected animals. A two-sided Welch *t*-test was performed to compare the different groups. ANOVA and, if significant, Tukey post hoc tests were performed to compare the different time points. The data of the boxplots are described in detail in Table A2. Scale bars (**A**,**B**): 100 μm. Abbreviations: M: monastral blue.

**Figure 3 ijms-24-13893-f003:**
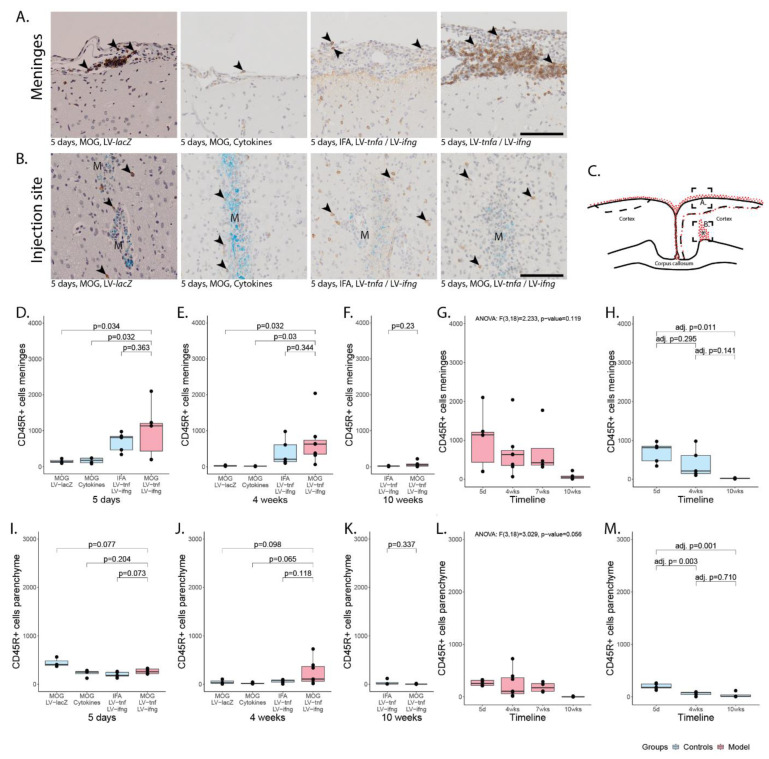
Meningeal and parenchymal infiltration with CD45R-positive B-cells. (**A**) Representative images of CD45R+ B-cells in meninges (arrowheads) and (**B**) in parenchyma (arrowheads). (**C**) Representative illustration of the inflammation pattern (red dots) five days after intracortical injection (asterisk), and delineation of the area of which images in A and B were taken (dotted squares). (**D**–**F**) Quantification of CD45R+ T-cells detected in meninges for all study groups five days, four weeks, or ten weeks after lentiviral injection. (**G**) Timeline of data presented in (**D**–**F**) of MOG-immunized, LV-*tnfa*- and LV-*ifng*-injected animals and (**H**) of non-immunized, LV-*tnfa*- and LV-*ifng*-injected animals. (**I**–**K**) Quantification of CD45R+ T-cells detected in parenchyma for all study groups five days, four weeks, or ten weeks after lentiviral injection. (**L**) Timeline of data presented in (**I**–**K**) of MOG-immunized, LV-*tnfa*- and LV-*ifng*-injected animals and (**M**) of non-immunized, LV-*tnfa*- and LV-*ifng*-injected animals. A two-sided Welch *t*-test was performed to compare different groups. ANOVA and, if significant, Tukey post hoc tests were performed to compare the different time points. The data of the boxplots are described in detail in Table A3. Scale bars (**A**,**B**): 100 μm. Abbreviations: M: monastral blue.

**Figure 4 ijms-24-13893-f004:**
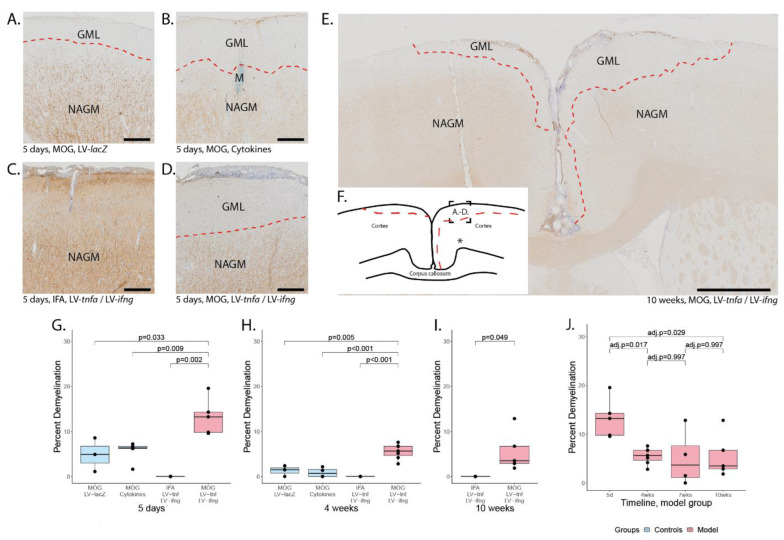
Subpial demyelination. Representative images of anti-MBP stainings demonstrating the subpial demyelination five days after the intracerebral injection of (**A**) LV-*LacZ*, (**B**) recombinant protein TNFα and IFNγ, (**C**) LV-*tnfa*/LV-*ifng* in non-immunized animals and (**D**) LV-*tnfa*/LV-*ifng* in MOG-immunized animals. (**E**) Overview image representing the subpial demyelination observed 10 weeks after the intracerebral injection. (**F**) Representative illustration of observed cortical demyelination pattern (dotted lines) 10 weeks after intracortical injection (asterisk). Dotted square shows the localization of the images in (**A**–**D**). (**G**–**I**) Quantification of demyelinated area: Boxplots show the percent of demyelinated area across both hemispheres for all study groups, five days, four weeks, or ten weeks after lentiviral injection. A two-sided Welch *t*-test was performed to compare between groups. (**J**) Timeline of data presented in (**G**–**I**) of the MOG-immunized, LV-*tnfa*/LV-*ifng*-injected animals. An ANOVA and Tukey post hoc test were performed to compare the different time points. The data of the boxplots are described in detail in Table A5. Scale bars (**A**–**D**): 100 μm, (**E**): 500 μm. Abbreviations: GML: grey matter lesion; NAGM: normal-appearing grey matter.

## Data Availability

All data analyzed during this study are included in the published article. All scans from the histological sections are deposited at the BioImage Archive of the BioStudies database under the accession number S-BIAD884 (European Molecular Biology Laboratory - European Bioinformatics Institute (EMBL-EBI), Wellcome Genome Campus, Hinxton, Cambridgeshire, CB10 1SD, UK. https://www.ebi.ac.uk/biostudies/, accessed on 7 September 2023).

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
