# Peer review of "An Animal Model for Chronic Meningeal Inflammation and Inflammatory Demyelination of the Cerebral Cortex"

_ijms, 2023, doi:10.3390/ijms241813893_

Round 1
Reviewer 1 Report
The manuscript presented by Enz et al. has proposed an improved animal model that could generate long-lasting meningeal inflammation and inflammatory demyelination of the cerebral cortex. The scope of this work fits the journal and the topic of the special issue well. The authors have also discussed the strength and weakness of the current model compared with some previous studies, giving additional knowledge regarding necessary factors in maintaining chronic meningeal inflammation and inflammatory demyelination. The overall quality of this work is good but a few issues need to be handled, which are listed below:
Major:
1. Section 2.1, figure 1B: In the authors’ experiment design, the time point for analysis has been set to 5 days, 4 weeks, 7 weeks, and 10 weeks, but the final results indicated that only experimental model (MOG/IFA, LV-tnfa/LV-ifng) has data at all 4 time points. Ideally and strictly speaking, it is highly suggested that the authors should provide sufficient data for all experimental groups at all 4 time points. The authors may want to provide additional statements to explain why some groups are not analyzed at those time points, if the authors insist to use the current experiment design scheme. In addition, the authors did not mention the time point, 7 weeks, in their section 2.1 when figure 1B has been mentioned. Please make sure the authors’ arguments are coincident with the figures. If the authors decide to remove the time point, 7 weeks, please renew the figures accordingly, as I notice that most discussions are about 5 days, 4 weeks, and 10 weeks.
2. Figure 1 D&E: It is highly suggested that the authors should provide enlarged images of neuronal and glial cell morphology for figure 1 D&E. The current version may be challenging to observe morphologies carefully.
3. Section 3 Discussion, 9th paragraph, line 412-413, “which both have been shown to lead to oligodendrocyte death and demyelination”: Though this correlation has been observed in other studies, this statement may still not be suitable here, because the connection between overexpression of TNFa and INFr and death of oligodendrocyte is not clearly shown in the authors’ work. Citing this additional information here may cause confusion. If needed, the authors could relocate this information to previous paragraph where they discussed oligodendrocyte results, such as line 389-391 (“We speculate that in the initial phase…”), to support their speculation.
Minor:
1. Please remove “Title:” in the title. Clearly, there is a small format issue.
2. The entire manuscript frequently displays format errors in reference citations, such as “Error! Reference source not found”. It seems that this type of error is related to figure number, happening in sections 2.1, 2.6, and 2.7. Please correct it.
3. Many cited/mentioned figures in section 2.3 and 2.4 are missing figure numbers.
4. Figure 2&3: In the captions of figure 2&3, there is a sentence clearly copied from other figure’s caption,” which images in A-D were taken (dotted square)”, which is not coincident with the actual figures shown in figure 2&3.
Reviewer 2 Report
Please revise the paper by addressing the following issues:
1. Abstract: The background in the beginning of abstract. It should give two-to-three-line maximum background which can help let the readers easily understand what is the issue in the current research. Likewise, the purpose of the study in the abstract have simply given. I suggest to only mention the main variables/purpose of the study. A one to two lines recommendation at the end of the abstract need to be provided. I suggest to arrange the abstract more logically as it simply established.
2. The introduction begins with background however there are several statements without the support of references. I suggest to support these statements with references. Add recently published papers in the introduction and if possible remove the old papers. This will help shows the research gap newer. I suggest to better highlight the research innovations and gap in introduction. Why this study is necessary? What policy level problem this study is addressing? How the study is expected to provide any solution to that problem? How the choice of sample is complementing that problem? Are the results and policies generalizable? Introduction is silent in all these aspects.
3. Highlight the problem statement and clearly relate to the research questions.
4. Would you please reason both the novelty and the relevance of your paper goals?
5. The literature review section is missing. Please critically review the recent literature and describe the research gap more precisely at the end of the Literature Review section.
6. Structure of the manuscript: The current study is poor for the following reasons: (i) Insufficient or inappropriate paragraphing and (ii) the order of presentation is inappropriate. The Material and Methods section should be before the Results and Discussion section.
7. A detail discussion of results with reference to existing literature is required.
8. The conclusion of the study is missing. Please include some practical implications of your study findings in the conclusion.
9. References are to short please add more references. References also require some revisions for uniformity in pattern according to the style recommended by the Journal.
10. Please proof read the manuscript before submitting the revision.
Round 2
Reviewer 1 Report
The authors have sufficiently addressed my concerns. I have no further comments.
Reviewer 2 Report
In their replies, the authors have, in my opinion, satisfactorily addressed the issues raised by me. I feel pleasure to recommend the manuscript for publication.